# A bioavailable strontium ($^{87}Sr/^{86}Sr$) isoscape for Aotearoa New Zealand: Implications for food forensics and biosecurity

R. T. Kramer[1]*, R. L. Kinaston[1,2], P. W. Holder[3], K. F. Armstrong[3], C. L. King[1], W. D. K. Sipple[4], A. P. Martin[5], G. Pradel[6], R. E. Turnbull[5], K. M. Rogers[6], M. Reid[7,8], D. Barr[7], K. G. Wijenayake[8], H. R. Buckley[1], C. H. Stirling[7,9], C. P. Bataille[10]*

**1** Department of Anatomy, University of Otago, Dunedin, Aotearoa New Zealand, **2** BioArch South, Waitati, Aotearoa New Zealand, **3** Bio-Protection Research Centre, Lincoln University, Canterbury, Aotearoa New Zealand, **4** Department of Anthropology, California State University, Chico, California, United States of America, **5** GNS Science, Dunedin, Aotearoa New Zealand, **6** GNS Science, Lower Hutt, Aotearoa New Zealand, **7** Centre for Trace Element Analysis, University of Otago, Dunedin, Aotearoa New Zealand, **8** Department of Chemistry, University of Otago, Dunedin, Aotearoa New Zealand, **9** Department of Geology, University of Otago, Dunedin, Aotearoa New Zealand, **10** Department of Earth and Environmental Sciences, University of Ottawa, Ontario, Canada

* rtkramer92@gmail.com (RTK); cbataill@uOttawa.ca (CPB)

**Data Availability Statement:** All relevant data are within the paper and its Supporting Information files.

## Abstract

As people, animals and materials are transported across increasingly large distances in a globalized world, threats to our biosecurity and food security are rising. Aotearoa New Zealand is an island nation with many endemic species, a strong local agricultural industry, and a need to protect these from pest threats, as well as the economy from fraudulent commodities. Mitigation of such threats is much more effective if their origins and pathways for entry are understood. We propose that this may be addressed in Aotearoa using strontium isotope analysis of both pests and products. Bioavailable radiogenic isotopes of strontium are ubiquitous markers of provenance that are increasingly used to trace the origin of animals and plants as well as products, but currently a baseline map across Aotearoa is lacking, preventing use of this technique. Here, we have improved an existing methodology to develop a regional bioavailable strontium isoscape using the best available geospatial datasets for Aotearoa. The isoscape explains 53% of the variation ($R^2 = 0.53$ and RMSE = 0.00098) across the region, for which the primary drivers are the underlying geology, soil pH, and aerosol deposition (dust and sea salt). We tested the potential of this model to determine the origin of cow milk produced across Aotearoa. Predictions for cow milk (n = 33) highlighted all potential origin locations that share similar $^{87}Sr/^{86}Sr$ values, with the closest predictions averaging 7.05 km away from their true place of origin. These results demonstrate that this bioavailable strontium isoscape is effective for tracing locally produced agricultural products in Aotearoa. Accordingly, it could be used to certify the origin of Aotearoa's products, while also helping to determine if new pest detections were of locally breeding populations or not, or to raise awareness of imported illegal agricultural products.

**Funding:** CLK received funding from the Royal Society of New Zealand Marsden Fund Fast-Start Grant (Award #: 17-UOO-149): https://www.royalsociety.org.nz/. KFA and PWH received funding from Better Border Biosecurity through a contract from The New Zealand Institute for Plant and Food Research Limited (Contract #: 37373): https://www.b3nz.org.nz/ and from the Plant Biosecurity Cooperative Research Centre, Australia (Contract #: PBCRC2111): http://legacy.pbcrc.com.au/. PWH received funding from the Tertiary Education Commission Centre of Research Excellence PhD funding to the Bio-Protection Research Centre: www.tec.govt.nz/funding RTK received funding from the following organizations: Centre for Global Migrations Postgraduate Research 2019 and 2020 Grants: https://www.otago.ac.nz/global-migrations/ Australasian Society for Human Biology Studentship: https://www.australasianhumanbiology.com/ Royal Society Te Apārangi Skinner Fund: https://www.royalsociety.org.nz/ University of Otago Doctoral Scholarship: https://www.otago.ac.nz/graduate-research/scholarships/phd/ Origin Inference from Geospatial Isotope Networks Grant (NSF Award ABI-1565128): https://www.nsf.gov/awardsearch/showAward?AWD_ID=1565128 All funders had no role in study design, data collection and analysis, decision to publish, or preparation of the manuscript.

**Competing interests:** The authors have declared that no competing interests exist.

## Introduction

Aotearoa New Zealand's (hereafter Aotearoa) local and export economy as well as its natural environment have been jeopardized by the introduction of foreign pests and associated diseases [1–5] as well as the selling of impure and fraudulent food products marketed as Aotearoa-made [6–9]. Specifically, Aotearoa's food industry is vulnerable to tampered, counterfeited, adulterated, and simulated food products [6,8]. Ensuring the purity and integrity of Aotearoa products protects a multi-billion-dollar food industry, with dairy ($16.6 billion NZD), wine ($1.75 billion NZD), and mānuka honey ($314 million NZD) being key agricultural products that are the most common fraudulent goods on the national and international markets [8]. Fraudulent and cheaper varieties of each product result in significant financial losses for the Aotearoa economy [8]. For mānuka honey, it is estimated that the annual recorded profit of $314 million NZD is four times inferior to what would be expected without counterfeit products diluting the market [8]. Consequently, the Treasury of the New Zealand Government estimates that the Ministry of Primary Industries (MPI) spends approximately $400 million NZD each year on biosecurity programs that assess, contain, and prevent foreign pests and fraudulent foods from entering and exiting the country [9]. Understanding the origins and pathways of entry is a key piece of information towards this effort but is often very difficult or impossible to provide.

Provenancing biological products and organisms in Aotearoa using biogeochemical analyses has been investigated for both food forensics and biosecurity. To date it has been restricted to use of the isotopes of light elements (typically $\delta^2H$, $\delta^{18}O$, $\delta^{15}N$, $\delta^{13}C$, and some others) and trace element concentrations, including for the verification of products such as milk powders [10,11], wine [12], fetal bovine serum [13], or for insect pest biosecurity breaches [2]. Currently in Aotearoa, private company *Oritain* offers origin verification for food and related products [14]. Crown Research Institutes *GNS Science* [15], *Bio-Protection Research Centre* at Lincoln University [16] and the *Centre for Trace Element Analysis* at the University of Otago [17] also perform provenancing consultancy but focus more on research and conservation. *Oritain* utilizes light element stable isotopes ($\delta^2H$, $\delta^{18}O$, $\delta^{15}N$, $\delta^{13}C$) and trace elements (Na, K, Zn, Fe, +35 others) to authenticate and provenance materials within the country and globally. These geochemical methods for provenancing are proven and successful predictors of geographic origin for a variety of materials [18–22], but they have limitations.

Of the light element stable isotopes, $\delta^2H$ and $\delta^{18}O$ found in biological tissues are primarily derived from regional precipitation, drinking water, and atmospheric diatomic oxygen [23,24]. Ehleringer et al. [24] found that $\delta^2H$ and $\delta^{18}O$ present in organic tissues displays "spatially explicit patterns" that vary predictably with geography including altitude, latitude, temperature, and continentality [19,25]. However, their interpretation is complex as these isotopes fractionate during biological and physical processes as they circulate through ecosystems and tissues [23,26]. Therefore, while specific tissues usually preserve the spatial patterns observed in precipitation, predicting them requires the impractical development of tissue-specific conversion equations to account for isotopic fractionation [27–29]. In addition, $\delta^2H$ and $\delta^{18}O$ are sensitive to evaporative losses and to exchange, which complicates sample collection, storage, and application [19,25,30–32]. Importantly, $\delta^2H$ and $\delta^{18}O$ variations on the landscape vary continuously and at low spatial resolution, as well as seasonally, making the values redundant regionally [33–35]. Although, in rare situations, the spatial and seasonal distributions of $\delta^2H$ and $\delta^{18}O$ can be useful to differentiate between materials originating from the northern and southern hemispheres [2]. Generally, however, all else being equal it can be challenging to isotopically distinguish between ecosystems that are geologically and climatically similar using only $\delta^2H$ and $\delta^{18}O$ [34,36] because the variation and patterning of atmospheric isotopes may

not be distinct enough. This issue is most important in Aotearoa and the greater Pacific where islands and small land masses do not experience the same continentality effects that lead to distinct isotopic variation in other regions of the world.

The other light isotopes, $\delta^{13}$C and $\delta^{15}$N, are primarily used to look at variations in diet for humans and animals and farming practices [22,37–42]. Singularly, these isotope tracers have limited application to geographical origin prediction but combining them with other isotopes can assist with provenancing efforts. This is because $\delta^{13}$C and $\delta^{15}$N vary predictably with the climate, dietary resources, and long-term land-use effects as they differentiate between $C_3$ and $C_4$ plants, protein consumption in the diet, and identifying the use of conventional and organic fertilizers in farming systems [37,40,43]. Lastly, all light isotopes are susceptible to variation introduced through the global supermarket, where nonlocal products may be incorporated into modern human diets and blur the geochemical signature within the sample [19,37,44,45].

In addition to the light stable isotopes used for provenancing, trace elements (chemical elements found in low concentrations of less than 100 parts per million (ppm) in the mineral matrix) are also not used to predict origins but can be used for samples of known origin to determine if they have similar or distinguishable chemical profiles from other, potentially fraudulent, samples [46,47]. Chemical profile comparisons can be based on just a few key elements or on a whole suite of elements depending on the sample and reference materials available [47]. For example, trace elements could identify fraudulent food products, like wine or tea, by comparing the chemical fingerprint of the suspicious sample to the authentic product [20,21]. The caveat is that this requires *a priori* knowledge about the chemical profile for all potential regions of origin to determine if the sample of interest classifies into a particular region or group [20]. While feasible, this would require an extensive database to store the chemical fingerprints for every type of reference material that may need to be provenanced. Furthermore, Koffman et al. [48] found that trace elements from Aotearoa sediments lack the regional variability observed when using lead, neodymium, and strontium isotopes, but trace elements have successfully been used to demonstrate regional variability in Aotearoa soils at multiple scales [49,50].

The analysis of the radiogenic strontium isotope ratio, $^{87}$Sr/$^{86}$Sr, is an alternative approach that has the potential to be broadly applicable without the need for situation-specific research and development. Internationally, this isotope system has been a key investigative method used to predict the region-of-origin for plants, animals, insects, and other biological materials [18,51–56]. Ecosphere $^{87}$Sr/$^{86}$Sr variation primarily reflects the underlying geology but also other processes including chemical weathering, alluvial and fluvial erosion, soil processes, and aerosol deposition (sea salt, volcanic ash, dust, loess) [51,55,57]. Strontium in the geosphere enters the tissues of biological organisms through their uptake of water and ingestion of dietary resources. This integrated $^{87}$Sr/$^{86}$Sr fraction is referred to as biologically available or "bioavailable". Any minor isotopic fractionation that occurs in the environment is corrected accordingly during analysis [51,52,55,58,59]. Therefore, biological tissues usually retain the "isotopic fingerprint" of the local ecosystem from which dietary resources were obtained [58]. Different biological tissues (plant leaves, hair, nails, teeth, and bone) form and remodel at different rates and the integrated $^{87}$Sr/$^{86}$Sr values in these tissues should reflect where organisms lived at different time frames of their lives [60–62].

Strontium isotope analysis as a tool in forensic tracing has received limited attention to date in Aotearoa mostly due to the cost of analysis when scaled to very large numbers of samples, and the lack of a bioavailable $^{87}$Sr/$^{86}$Sr baseline. Recent efforts have gone some way to tackling these issues, with technical advances to address the former [63] and with Duxfield et al. [64] aggregating published geological data to create a $^{87}$Sr/$^{86}$Sr baseline. The recent baseline [64]

summarized the variation in geological $^{87}Sr/^{86}Sr$ for Aotearoa and did not predict the bioavailable $^{87}Sr/^{86}Sr$ produced through the complex environmental system that $^{87}Sr/^{86}Sr$ cycles through. Duxfield et al. [64] used a case study to compare plant bioavailable $^{87}Sr/^{86}Sr$ values to their geologic baseline and found that the values fell within the expected ranges based on the underlying lithology but demonstrated smaller $^{87}Sr/^{86}Sr$ value ranges than the geological unit they were associated with. These results are most likely due to the geological $^{87}Sr/^{86}Sr$ baseline not considering the exogenous sources of $^{87}Sr/^{86}Sr$ in the biosphere that are influenced by climatic, atmospheric, and environmental conditions. Also, some of the published geological data they used are based on $^{87}Sr/^{86}Sr$ compositions for individual mineral phases rather than the bulk rock, which homogenizes mineral signatures.

Similar to baselines, isoscapes are models that predict the spatial distribution of isotopes by incorporating various sources that contribute to the bioavailable isotope "pool" of a region [65–72]. Strontium isoscapes have been produced at the continental scale for North America [73], Central America [74], Africa [75], and Europe [76], at regional scales for Australia [77], the Caribbean [78], China [79], the Netherlands [80], France [81], southwest Sweden [82], the Modena [83], and South Korea [84], as well as for island locales such as mainland UK [85] and Ireland [86]. For strontium isotopes, the leading approach to create regional and global isoscapes [51,56,87–89] uses bioavailable $^{87}Sr/^{86}Sr$ values of georeferenced samples (soil, plants, and small local organisms) and a machine-learning framework that considers contributions from the ecosphere to predict bioavailable $^{87}Sr/^{86}Sr$ values across the region of interest.

In this paper, we introduce the first bioavailable $^{87}Sr/^{86}Sr$ isoscape model for Aotearoa constructed using a machine-learning approach [51,88] and the best available dataset of geospatial predictors. Specifically, the isoscape construction uses a random forest (RF) model framework that considers a variety of climatic, atmospheric, and environmental variables that may contribute to the spatial distribution of bioavailable $^{87}Sr/^{86}Sr$ throughout Aotearoa. The RF model also uses bioavailable $^{87}Sr/^{86}Sr$ data from plants, soils, and animals living in Aotearoa to calibrate the model and ensure that the predicted $^{87}Sr/^{86}Sr$ model values reflect the actual $^{87}Sr/^{86}Sr$ values obtained from the real-world samples.

Bataille et al. [51] discuss the optimal substrate to sample (plants and soils versus small local animals) when constructing $^{87}Sr/^{86}Sr$ isoscapes and concluded that while local animals are preferred because they integrate multiple sources of bioavailable $^{87}Sr/^{86}Sr$, plants and soils are acceptable alternatives. It is, however, important to consider that plants of different rooting depths take up varying $^{87}Sr/^{86}Sr$ values depending on the exchangeable soil fraction (the fraction of topsoil sample that is extractable when leached using an ammonium nitrate solution) [51,90,91]. To compensate for this, the sampling strategy of this study targeted plants of varying root depths (shallow, medium, and deep) at each sampling location to ensure that any intra-site $^{87}Sr/^{86}Sr$ variability was captured. Shallow roots are defined as slender, branched, fibrous or creeping roots that grow close to the surface. Medium roots include larger plants whose roots penetrate one to two meters below the surface soil. Deep-rooted plants have tap roots that consist of a primary root that penetrates deep into the soil at depths greater than three meters. Using the RF-based methodology allowed us to explain the main environmental influences on the $^{87}Sr/^{86}Sr$ isoscape and we then validated the use of for provenancing using cow milk samples collected from farms across Aotearoa.

## Materials and methods

### Bioavailable $^{87}Sr/^{86}Sr$ sample distribution and descriptive statistics

Plant sample collection sites were chosen based on a strict set of inclusion criteria to circumvent potential anthropogenic and natural Sr contaminants. These criteria avoided farms,

pastures, drainage ditches, and other human-made earthworks, and being restricted to public lands with road-access. Riparian areas, land in flood areas and along watercourses, were avoided to avoid any mixing of exogenous Sr sources [92]. Also, the topographic relief and elevation of the regions limited where and when samples could be taken. These factors greatly limited the distribution of the samples (Fig 1). Potential sample stops were planned from an ArcGIS shapefile layer with a highway road feature class overlaid on a geological map to mark where there was a geological change within each area along the route. Leaves were collected from plants of each root-type (shallow, medium, and deep) located within a 5 m diameter of one another at each sample site (when available) to capture a higher resolution of $^{87}Sr/^{86}Sr$ variation for the single locality. Samples were stored in labeled, plastic sample bags and dehydrated later the same day in a clean SunBeam food dehydrator before being stored in new clean and labeled, plastic sample bags. Plant type, root-depths, and other metadata are provided in the Supplementary Information (S3 File).

Topsoil samples (0–20 cm) were collected by GNS Science in the Nelson, Otago, and Southland regions (Fig 1). Samples were collected by hand auger and the sub-2-mm portion was retained after sieving and drying at 40˚C [49,93]. Additional soil and plant $^{87}Sr/^{86}Sr$ data were provided by PWH and KFA for the Christchurch, Bay of Plenty, Auckland, and Northland regions. To date, very few modern local human and animal bioavailable $^{87}Sr/^{86}Sr$ data have been published for Aotearoa and only eight (two sheep and six humans) were available to include in this study (Fig 1).

Multiple analytical methods were utilized to analyze the plant and soil samples and generate the $^{87}Sr/^{86}Sr$ data. GNS topsoil (n = 71) and collected plant samples (n = 185) were prepared under the supervision of the lab technician in the Centre for Trace Element Analysis, University of Otago, Dunedin. GNS topsoil subsamples of 1g were leached in 2.5 mL of 1 M ammonium nitrate ($NH_4NO_3$) solution and agitated overnight to extract the bioavailable strontium fraction [53,81,92,94]. Then, the topsoil samples underwent Sr separation column procedures as described previously [76,95,96], while the collected plant samples utilized an automated ion-exchange chromatography method using a 3-ml column (ESI, Part number CF-MC-SrCa-3000) filled with DGA (diglycolamide) resin (TrisKem International, Bruz, France) as described by Wijenayake [97].

Strontium isotope measurement of the collected plant and GNS topsoil samples was conducted at the Centre for Trace Element Analysis, University of Otago using a Nu Plasma-HR MC ICP-MS (Nu Instruments Ltd., UK) following previously reported protocols [76,95–99]. Repeated measurement of the NIST SRM 987 strontium isotope reference material (sourced from the National Institute of Standards and Technology (NIST), USA) and the HPS in-house strontium isotope standard, that bracketed every six samples, were used to monitor the accuracy and external precision of the measurements. We obtained average values of 0.71025 ± 0.00002 (2 SD, n = 70) for NIST SRM 987 and 0.70761 ± 0.00009 (2 SD, n = 58) for HPS in very good agreement with previously reported values. Any instrumental mass fractionation present was corrected for using the exponential mass fractionation lay by normalization to $^{86}Sr/^{88}Sr = 0.1194$. Procedural blanks were run with each batch of 6 samples and all yielded negligible Sr levels of < 250 pg. The additional soil and plant samples (n = 126) utilized Sr separation column procedures as described by Pin & Bassin [100]. Sr isotope measurement of the soil samples from [101] was conducted using a Nu Plasma-HR MC ICP-MS (Nu Instruments Ltd., UK) at the Centre for Trace Element Analysis, University of Otago, Dunedin (standard = NIST SRM 987, n = 19, $^{87}Sr/^{86}Sr$ 0.710274 ±0.000023 (2 SD)); and the additional soil and plant samples [102] measured using Isotopx Phoenix TIMS (thermal ionization mass spectrometry) at the University of Adelaide (NIST SRM 987, n = 7, $^{87}Sr/^{86}Sr = 0.710245 ± 0.000008$ (2 SD).

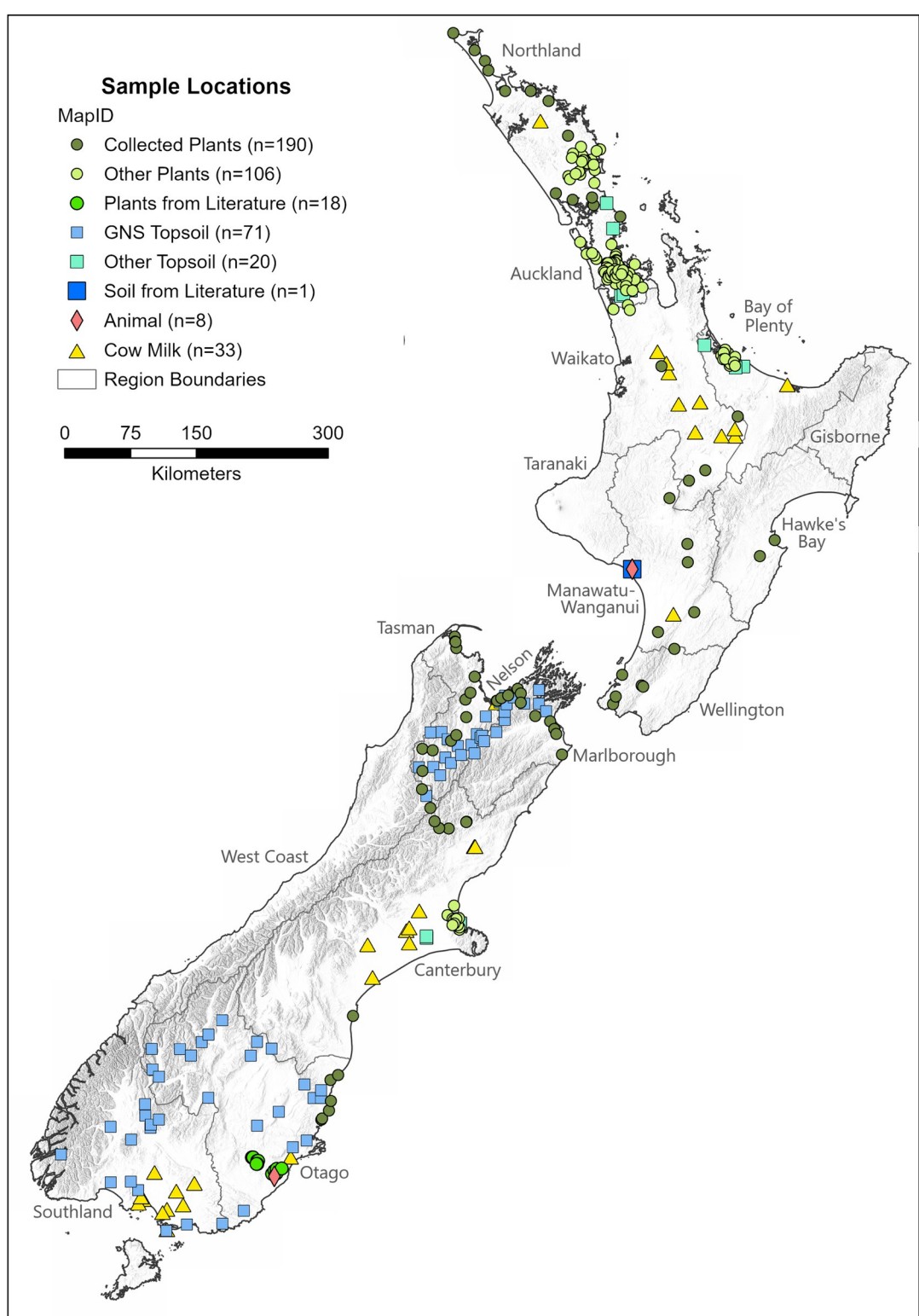

**Fig 1. Locations of samples used for developing and validating the bioavailable $^{87}$Sr/$^{86}$Sr isoscape.** Fig 1 developed in ArcGIS Pro and includes regional boundaries and coastlines feature layers (sourced from Natural Earth) and hillshade feature layer (created using GNS DEM 8m shapefile). Map projected to the New Zealand Transverse Mercator (NZTM 2000).

Further details regarding preparation and analysis for all plant and soil samples are available in the Supplementary Information (S1.1–5 in S1 File). Altogether, 414 bioavailable $^{87}$Sr/$^{86}$Sr data (Fig 1) were used to construct the strontium isoscape, specifically comprising data from 314 plants (182 unique locations), 92 topsoils (84 unique locations), and eight mammals (two unique locations).

## Bioavailable $^{87}$Sr/$^{86}$Sr isoscape

This study follows the methodology established by Bataille and colleagues [51,93] using a RF model framework. Model construction used bioavailable, geo-referenced $^{87}$Sr/$^{86}$Sr data gathered for Aotearoa, many geomatic auxiliary variables in the form of global rasters (gridded matrix of cells) used as covariates, and an existing process-based bedrock $^{87}$Sr/$^{86}$Sr model [87,88] to produce the final predicted $^{87}$Sr/$^{86}$Sr isoscape model. The RF R-script provided by Bataille et al. [51,88] optimizes the regression model with the root mean square error (RMSE) and uses five 10-fold repeated cross-validations. Additionally, the R-script includes using the *Variable Selection Under Random Forest* (*VSURF*) package [103] to identify relevant and highly predictive variables [51,87–89]. The relationships between the variables selected by the *VSURF* function and the bioavailable $^{87}$Sr/$^{86}$Sr variability were assessed using variable importance purity measure and partial dependence plots [51]. The auxiliary variables were obtained from various sources summarized in S2 Table 1 in S2 File and represent geological, climatic, and environmental variables that may influence bioavailable $^{87}$Sr/$^{86}$Sr variability. Most variables were available as global rasters that were trimmed to an Aotearoa extent and projected to the New Zealand Transverse Mercator 2000 coordinate system.

To assess the uncertainty of the isoscape, we used quantile RF regression model to generate quartile-1 and quartile-3 models using the log-transformed $^{87}$Sr/$^{86}$Sr values. We use a log-transformation because the $^{87}$Sr/$^{86}$Sr data have a positively skewed distribution. Then, the quartile-1 model was subtracted from the quartile-3 model (i.e., Q3-Q1) to generate a final interquartile range raster at a resolution of 1 km.

## Cow milk samples

Dairy farms throughout the country that contributed cow milk samples agreed to a controlled feeding regime where all cattle were expected to be pasture-fed on-site and were not provided with supplementary feed options [97]. Therefore, the assumption was that the $^{87}$Sr/$^{86}$Sr values of the cow milk would reflect the underlying geology and local atmospheric conditions with no contamination from exogenous $^{87}$Sr/$^{86}$Sr sources. Details regarding the isotopic preparation and analytical methods for the cow milk samples are provided in the Supplementary Information (S1.5 in S1 File).

To assess the validity of the bioavailable $^{87}$Sr/$^{86}$Sr isoscape developed, we predict the region-of-origin for cow milk samples and then assess the accuracy of that assignment relative to their actual origin. Firstly, origin predictions are computed using *assignR* [104], which operates in a semi-parametric Bayesian framework to calculate the posterior probability of the sample belonging to each cell within the isoscape raster. Statistically, *assignR* uses a maximum likelihood assignment model and employs Bayes Theorem to calculate the probability that a sample originates from a geographic location given the isotopic signature of the sample [105]. The statistical methodology of the *assignR* code [104] is adapted from Wunder [71] and Vander Zanden et al. [106]. To facilitate visualization, we created maps displaying the top 33% by area of the predicted surface for each sample (SI4) with the top 33% areas coded as 1 and other areas coded as 0. Once the top 33% probability maps are produced, they are brought into ArcGIS Pro. Then, we calculated the accuracy of the nearest predicted geographic origin (part of the

top 33% probability) for each sample in terms of how close they were in km to the actual place of origin for the cow, using the Measure Distance tool in ArcGIS Pro. We also measured the distance from the nearest predicted cell for the top 20% and top 10% probability surfaces to assess the trade-offs between accuracy and precision for these arbitrary thresholds. We do not provide the maps for the top 20% and 10% predictions, but they can be provided upon request.

## Results

### $^{87}Sr/^{86}Sr$ isotopes and Aotearoa bioavailable $^{87}Sr/^{86}Sr$ isoscape

The bioavailable $^{87}Sr/^{86}Sr$ data for the plant and soil samples are available in the Supplementary Information (S3 File). The combined values for plants, soils, and local mammals demonstrate comparable distributions, with Quartile 1 = 0.70738 ± 0.00002, median = 0.70832 ± 0.00003, and Quartile 3 = 0.70897 ± 0.00108 (Fig 2). The animal substrate samples display a tighter range of $^{87}Sr/^{86}Sr$ values compared to the plants and topsoils, with soil displaying the largest variability in $^{87}Sr/^{86}Sr$, but 90% of all data fall within values of 0.70500 and 0.71250 (Fig 2).

The final $^{87}Sr/^{86}Sr$ isoscape for Aotearoa uses the best performing RF model that considered all auxiliary variables (S2 Table 1 in S2 File) and 414 bioavailable $^{87}Sr/^{86}Sr$ values (S3 File). The RF regression model produces a map demonstrating the mean $^{87}Sr/^{86}Sr$ prediction ($R^2$ = 0.53, RMSE = 0.00098) ranging from 0.70567 to 0.71118, for the entire country including the Chatham Islands (Fig 3A). The accompanying interquartile range raster (Fig 3B) demonstrates a standard error ranging from 0.0001 to 0.002 for the country.

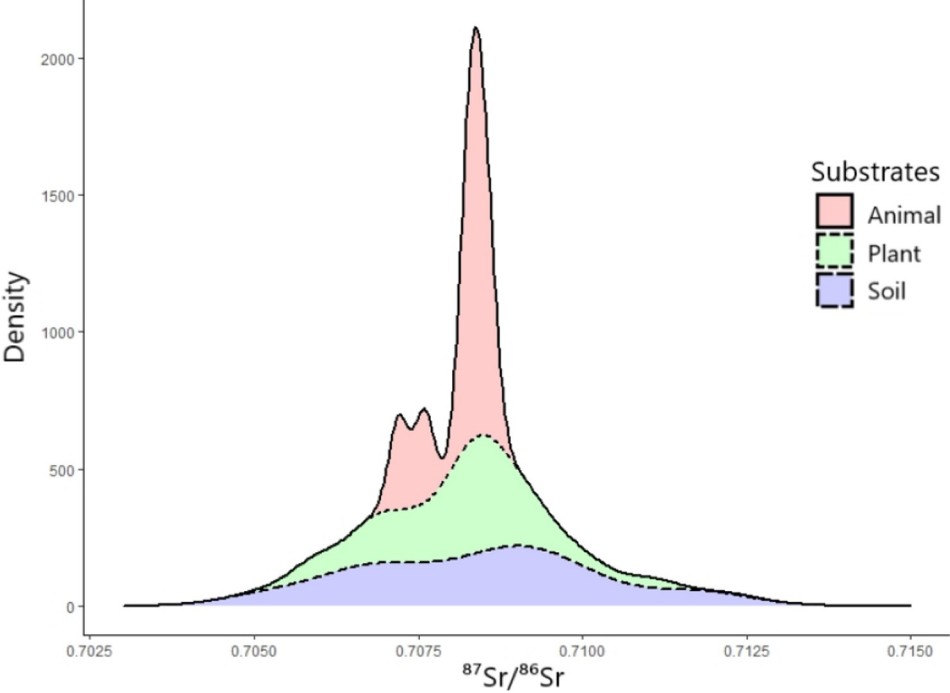

**Fig 2. Stacked plot illustrating the distribution of the collected $^{87}Sr/^{86}Sr$ variability by substrate.** The animal substrate (n = 8) samples in pink display a tighter range of $^{87}Sr/^{86}Sr$ values compared to the plants in green (n = 314) and topsoils in blue (n = 92) samples that are all used to construct and calibrate the bioavailable $^{87}Sr/^{86}Sr$ isoscape model.

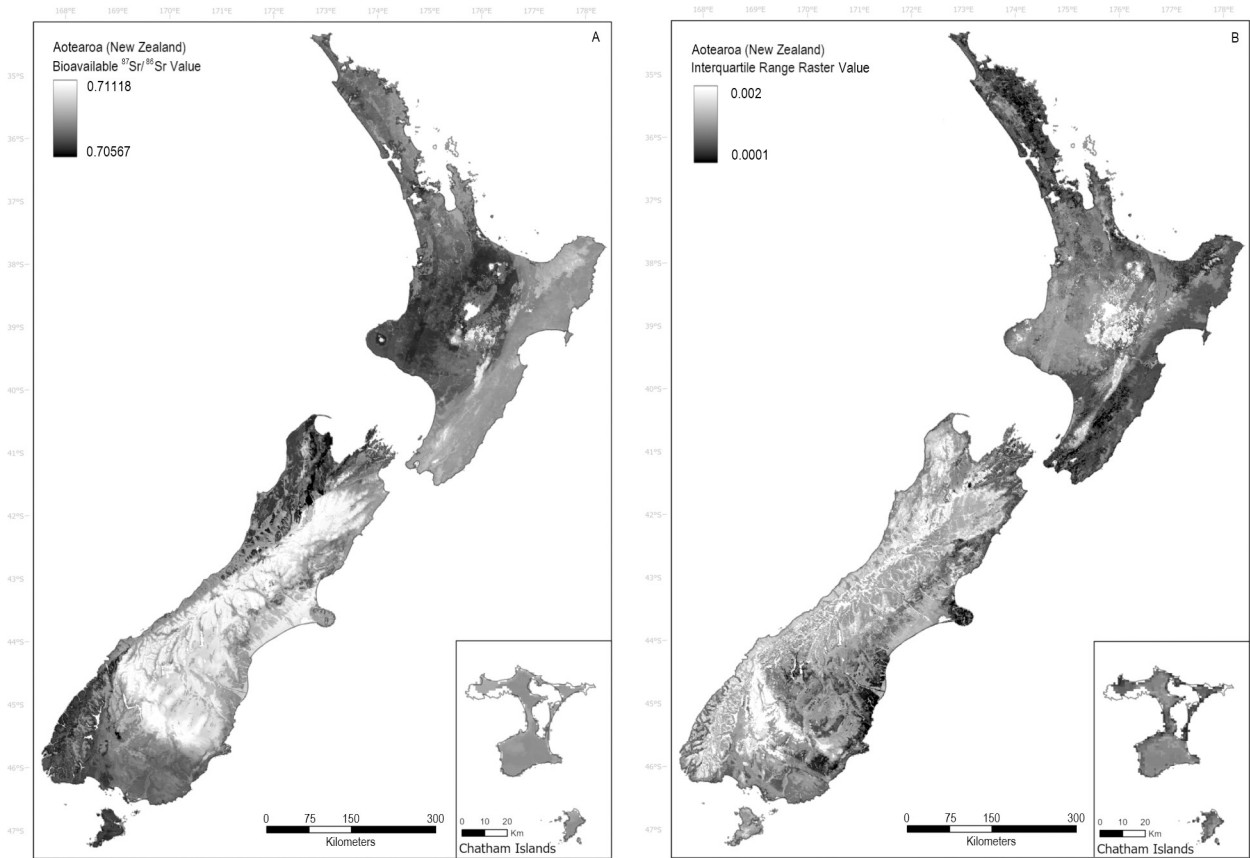

**Fig 3.** Bioavailable $^{87}$Sr/$^{86}$Sr isoscape (A) and interquartile range raster (B) for Aotearoa created using random forest regression. The bioavailable $^{87}$Sr/$^{86}$Sr isoscape (A) (R$^2$ = 0.53, RMSE = 0.00098) demonstrates the predicted $^{87}$Sr/$^{86}$Sr values, ranging from 0.70567 to 0.71118, for the entire country including the Chatham Islands. Log-transformed $^{87}$Sr/$^{86}$Sr values of the bioavailable data were used to construct quartile-1 and quartile-3 regression models that were then subtracted from one another (Q3-Q1) to create the final interquartile range raster (B) with values ranging from 0.0001 to 0.002. Figure developed in ArcGIS Pro with a coastlines feature layer (sourced from Natural Earth) and projected to NZTM 2000.

## Assessment of Aotearoa $^{87}$Sr/$^{86}$Sr model performance

The *VSURF* package selected 11 predictive variables (Fig 4A). These include dust and sea salt aerosol deposition (r.dust, r.ssa, r.ssaw), geological attributes (r.age, r.toprock, r.GNSagemax), temperature (r.mat), elevation (r.elevation), and soil characteristics (r.ph, r.clay, r.nitrogen). Initially, the *VSURF* selected 10 features, but when r.nitrogen was forcibly added to the RF function, the amount of variation explained increased by ~2%. Therefore, r.nitrogen was included in the final model regression. The dominant predictive variables are the rock type classification (r.toprock) and soil pH in H$_2$O solution (r.ph) based on their Percent Increased Mean Squared Error (%IncMSE) values (Fig 4A). The higher the %IncMSE value, the more important the predictor variable [107]. The n-fold cross validation explains 53% of the variation in the bioavailable $^{87}$Sr/$^{86}$Sr model, with an RMSE of 0.00098 over the dataset (Fig 4B). The uncertainty appears uniform across the prediction range (Fig 4B).

Partial dependence plots are used to examine the association between the predictors and bioavailable $^{87}$Sr/$^{86}$Sr (Fig 5). These illustrate that $^{87}$Sr/$^{86}$Sr values increase with increasing soil pH (r.ph) and elevation (r.elevation). On the other hand, inverse relationships are apparent where $^{87}$Sr/$^{86}$Sr values increase with decreasing mean annual temperature (r.mat), principal surface lithology type (r.toprock), rock age (r.age, r.GNSagemax), and clay content (r.clay).

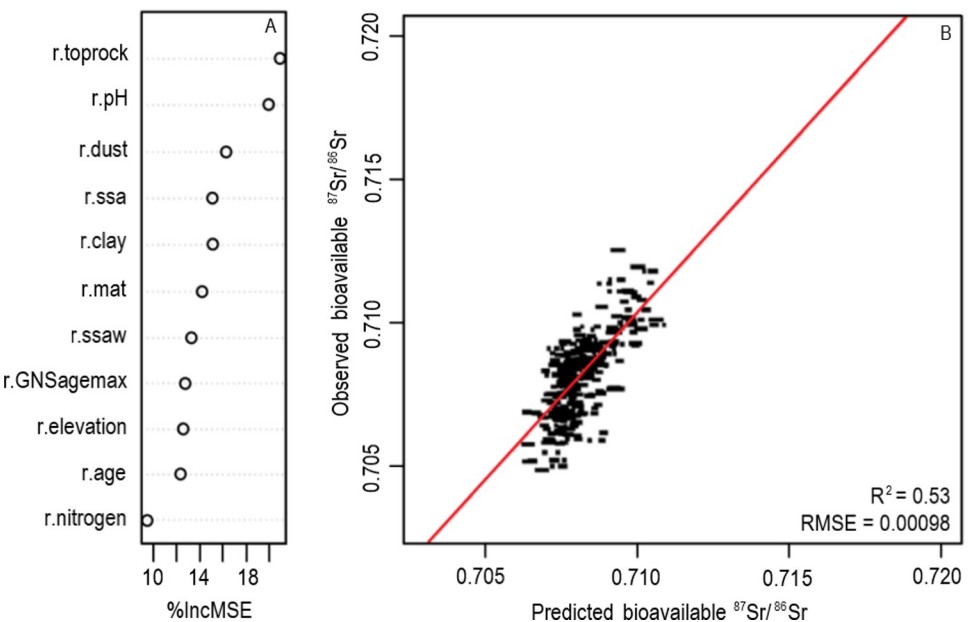

**Fig 4. Random forest regression performance plots.** Plots demonstrate (A) model variable importance plot and (B) n-fold cross validation with best fit line in red.

The "toprock" raster (Fig 5) consists of 67 different lithological classifications (detailed in S2 Table 2 in S2 File) where the first few categories have the highest $^{87}$Sr/$^{86}$Sr values: 1 = floodplain alluvium; 2 = volcanic ashes older than Taupō pumice; 3 = Loess; 4 = Sandstone;

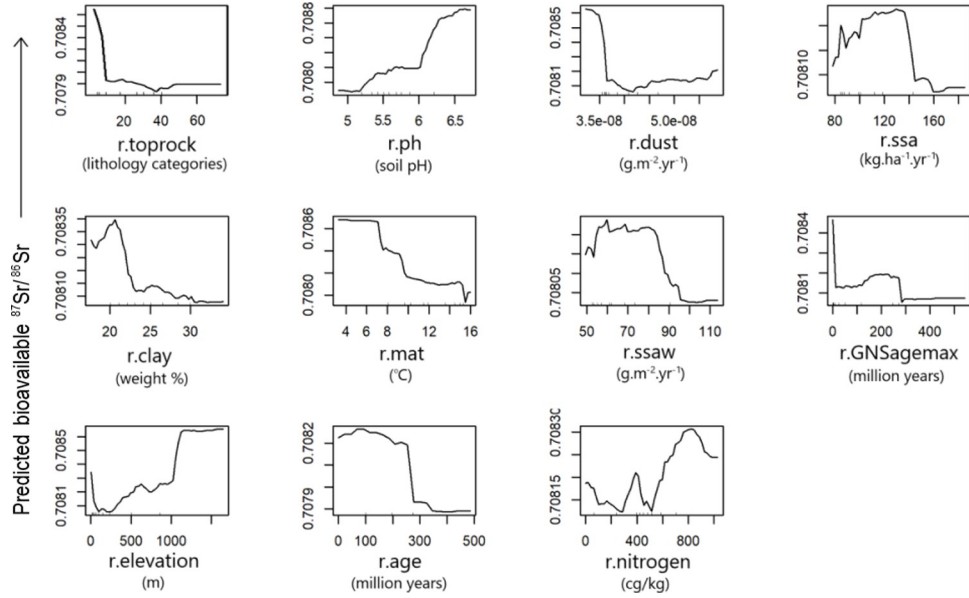

**Fig 5. Partial dependence plots with predictive variables (x-axis) and predicted bioavailable $^{87}$Sr/$^{86}$Sr (y-axis).** Units of measurement are provided on the x-axis for each variable. For r.toprock, the x-axis represents the numbered categories of rock type (S2 Table 2 in S2 File and S2 Fig 1 in S2 File) obtained from the New Zealand Land Resource Information System online portal. Refer to the Supplementary Information, S2 Table 1 in S2 File, for descriptions of all variables used in the RF model and their sources.

5 = Greywacke; etc. Because this variable is categorical, its correlation with $^{87}Sr/^{86}Sr$ is most likely coincidental. Both rock age variables, r.GNSagemax and r.age, (Fig 5) demonstrate a threshold-based relationship with bioavailable $^{87}Sr/^{86}Sr$ values showing little influence of age above 300 million years. This is consistent with previous findings demonstrating that the age of rocks is a major contributor to bioavailable $^{87}Sr/^{86}Sr$ values [73,87,88]. The remaining predictive variables, dust and sea salt aerosol deposition (r.dust, r.ssa, r.ssaw) and nitrogen content (r.nitrogen), display non-linear relationships with the $^{87}Sr/^{86}Sr$ values.

## Model validation using cow milk

Generally, we found that cow milk with $^{87}Sr/^{86}Sr$ values near 0.70800 were difficult to assign because most of the isoscape values across Aotearoa fall in a narrow band between the range of 0.70750 and 0.70930. When a sample is within this interval, the probability surfaces show broad regions with high probabilities of origin corresponding to low assignment precision. Conversely, the model performs well when predicting origin for samples that are below 0.70750 or above 0.70930 because these values are much less common on the Aotearoa isoscape allowing for better precision. We produced prediction maps showing the top 33% of isoscape raster cells with the highest posterior probability for each cow milk sample (n = 33) available in the Supplementary Information (S4 File). The cow milk $^{87}Sr/^{86}Sr$ and the predicted $^{87}Sr/^{86}Sr$ from the isoscape show a good correlation ($R^2$ = 0.52) when compared with one another (Fig 6). When error bars are included, all cow samples fall within the 95% confidence interval of the linear regression line on the actual versus predicted bioavailable $^{87}Sr/^{86}Sr$ value plot (Fig 6), except sample Cow 7.

The maximum likelihood assignment model was quite accurate and all but one milk sample had high probability cells (top 33%) located within proximity (average of 7.05 kilometers) to

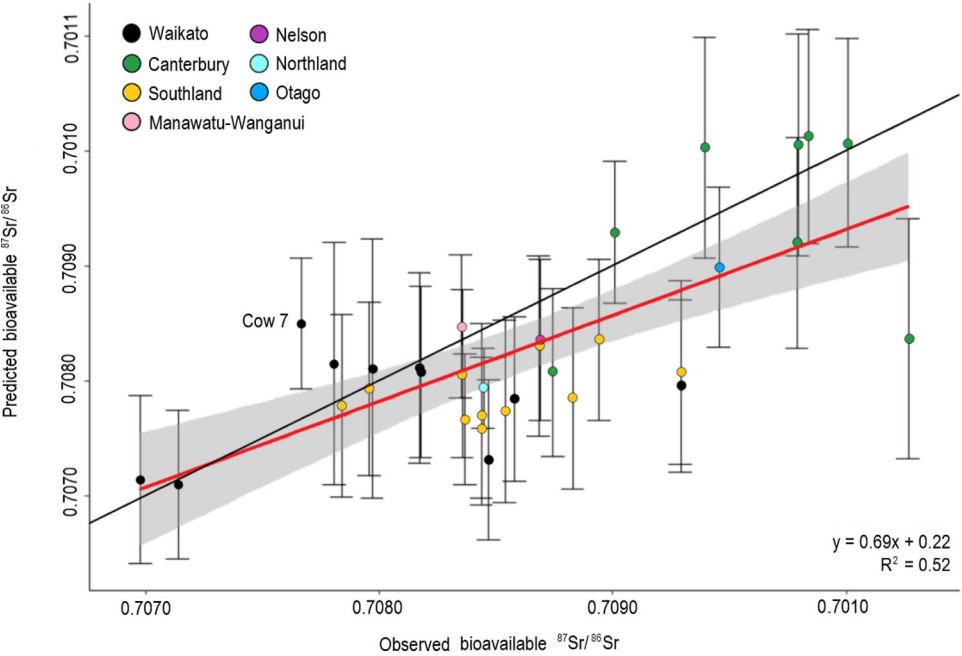

**Fig 6. Observed cow milk $^{87}Sr/^{86}Sr$ (x-axis) versus predicted $^{87}Sr/^{86}Sr$ values (y-axis) from the bioavailable isoscape.** Error bar values were extracted from the uncertainty raster for each cow milk sample location. The red line represents the line of best fit and the shaded region represents the 95% confidence interval (CI). The black line shows the 1:1 line between predicted and observed $^{87}Sr/^{86}Sr$ values. Sample points are color-coded by region (regions depicted in Fig 1). All cow milk samples fall within the 95% CI, except sample Cow 7 labeled on plot.

**Table 1. Cow milk sample data.**

| Sample ID | Latitude | Longitude | Region | Distance from known origin to nearest predicted cell on the probability surface (km) | | |
|---|---|---|---|---|---|---|
| | | | | Top 33% | Top 20% | Top 10% |
| Cow 1 | -38.512 | 176.171 | Waikato | 7.06 | 7.536 | 27.18 |
| Cow 2 | -37.883 | 175.462 | Waikato | 6.42 | 24.29 | 74.18 |
| Cow 3 | -37.783 | 175.425 | Waikato | 8.39 | 24.62 | 29.82 |
| Cow 4 | -38.483 | 175.826 | Waikato | 0 | 0 | 0.38 |
| Cow 5 | -38.173 | 175.877 | Waikato | 0 | 0 | 0 |
| Cow 6 | -38.201 | 175.600 | Waikato | 16.36 | 16.36 | 52.93 |
| Cow 7 | -37.963 | 177.000 | Waikato | 16.84 | 24.74 | 25.59 |
| Cow 8 | -37.669 | 175.308 | Waikato | 0 | 2.463 | 15.01 |
| Cow 9 | -38.436 | 176.341 | Waikato | 0.64 | 16.66 | 36.8 |
| Cow 10 | -38.510 | 176.348 | Waikato | 13.95 | 38.23 | 38.28 |
| Cow 11 | -43.623 | 171.976 | Canterbury | 0.15 | 1.32 | 20.05 |
| Cow 12 | -44.097 | 171.488 | Canterbury | 0 | 2.86 | 15.73 |
| Cow 13 | -42.767 | 172.936 | Canterbury | 1.56 | 2.89 | 7.689 |
| Cow 14 | -43.422 | 172.162 | Canterbury | 1.53 | 1.374 | 3.659 |
| Cow 15 | -42.767 | 172.949 | Canterbury | 0.47 | 0.45 | 0.43 |
| Cow 16 | -43.750 | 172.017 | Canterbury | 0 | 0 | 20 |
| Cow 17 | -43.595 | 172.017 | Canterbury | 0.58 | 1.15 | 24.78 |
| Cow 18 | -43.762 | 171.427 | Canterbury | 0 | 0 | 41.42 |
| Cow 19 | -46.598 | 168.363 | Southland | 4.86 | 112.76 | 112.76 |
| Cow 20 | -46.598 | 168.363 | Southland | 2.15 | 2.162 | 5.252 |
| Cow 21 | -46.390 | 168.380 | Southland | 3.16 | 3.306 | 5.331 |
| Cow 22 | -46.210 | 168.533 | Southland | 0 | 0.47 | 5.401 |
| Cow 23 | -46.006 | 168.229 | Southland | 0 | 0 | 8.51 |
| Cow 24 | -46.354 | 168.622 | Southland | 18.95 | 21.7 | 21.89 |
| Cow 25 | -46.140 | 168.805 | Southland | 12.87 | 13.32 | 33.52 |
| Cow 26 | -46.276 | 168.042 | Southland | 3.53 | 6.496 | 69.26 |
| Cow 27 | -46.310 | 167.965 | Southland | 2.91 | 96.34 | 141.2 |
| Cow 28 | -46.247 | 168.010 | Southland | 8.87 | 9.27 | 10.13 |
| Cow 29 | -46.421 | 168.313 | Southland | 86.24 | 114.08 | 123.52 |
| Cow 30 | -45.917 | 170.237 | Otago | 0 | 0 | 4.343 |
| Cow 31 | -35.323 | 173.772 | Northland | 1.24 | 1.203 | 1.232 |
| Cow 32 | -40.355 | 175.610 | Manawatu- Wanganui | 0 | 0 | 0 |
| Cow 33 | -41.293 | 173.238 | Nelson | 13.77 | 14.47 | 37.96 |

the actual dairy farm of origin for each represented region (Tables 1 and 2). We also measured the distance between the nearest probability cell and the dairy farm of actual origin for the top 20% and top 10% probability surfaces to assess the trade-off between precision and accuracy when different probability thresholds are used (Tables 1 and 2). While the accuracy of most origin predictions is high using the top 33% probability threshold, the precision is lacking for most samples. Conversely, the top 20% and top 10% probability thresholds produce precise origin predictions (i.e., they predict fewer regions of potential origin), but the accuracy (distance to known origin) decreases (Tables 1 and 2).

Cow milk samples originating from the Canterbury region produce the most accurate predictions (Tables 1 and 2). For all Canterbury cow milk samples, the prediction maps have high probability cells within 0 to 1.56 km of the actual location. The Waikato region milk samples

**Table 2. Descriptive statistics for cow milk top 33%, 20%, & 10% quantile predictions.**

|  | Region | Number | Average (km) | Minimum (km) | Maximum (km) | Standard Deviation (km) |
|---|---|---|---|---|---|---|
| **Top 33%** | Canterbury | 8 | 0.54 | 0 | 1.56 | 0.66 |
|  | Waikato | 10 | 6.97 | 0 | 16.84 | 6.85 |
|  | Southland | 11 | 13.05 | 0 | 86.24 | 24.95 |
|  | All Regions | 33 | 7.05 | 0 | 86.24 | 15.38 |
| **Top 20%** | Canterbury | 8 | 1.26 | 0 | 2.89 | 1.14 |
|  | Waikato | 10 | 15.49 | 0 | 38.23 | 12.82 |
|  | Southland | 11 | 35.54 | 0 | 114.08 | 47.63 |
|  | All Regions | 33 | 16.99 | 0 | 114.08 | 30.82 |
| **Top 10%** | Canterbury | 8 | 16.72 | 0.43 | 41.42 | 13.18 |
|  | Waikato | 10 | 30.02 | 0 | 74.18 | 22.66 |
|  | Southland | 11 | 48.80 | 5.25 | 141.20 | 53.27 |
|  | All Regions | 33 | 30.73 | 0 | 141.20 | 36.20 |

produce highly and moderately accurate predictions with a 6.97 km average distance away from the nearest predicted cell (Table 2). Samples Cow 4, 5, 8, and 9 are highly accurate with predictions 0–0.64 km from their actual locations. The remaining samples in the Waikato region have predictions 6.42 to 16.84 km away from their known locations (Table 2). For the Southland region, all cow milk samples originated near the city of Invercargill and display variable prediction accuracies. Samples Cow 19–23 and 26–27 are highly accurate (less than 5 km away); Cow 24, 25, and 28 are moderately accurate (6 to 20 km away); and Cow 29 is the least accurate being 86.24 km away from the nearest predicted cell (Table 2).

## Discussion

### Bioavailable strontium isoscape for Aotearoa

Understanding the geology and climate allows improved modeling of Sr isotope cycles from the geosphere into the biosphere for provenancing materials. Accordingly, the covariates used in the RF model here account for geological and atmospheric conditions in Aotearoa (S2 Table 1 in S2 File). Seven of the eleven highest performing covariates selected by the *VSURF* package consisted of geological variables (Figs 4 and 5), while the remaining four were atmospheric variables showing the contribution of dust and sea salt aerosols and mean annual temperature to bioavailable $^{87}Sr/^{86}Sr$ values.

Of the geological variables, there are several features that can explain the observed range of $^{87}Sr/^{86}Sr$ values (0.70560–0.71120 ± 0.002, Fig 3) across Aotearoa. These values reflect the variable composition of bedrock for the country, which has a rich geological history [108] that has resulted in areas with distinct mineralogy and rock ages, both factors which strongly affect strontium and consequently bioavailable $^{87}Sr/^{86}Sr$ [109]. Aotearoa has a compositionally diverse basement geology with formation ages between c. 500–100 million years ago (Ma) (Fig 7). The modern-day distribution of geology reflects terrane accretion and c. 23 Ma of tectonism along the strike-slip system, the Alpine Fault, which bisects the country [108,110,111]. The Alpine Fault juxtaposes terranes which were once contiguous for hundreds of kilometers, resulting in rocks of terrane affinity being present at either end of the country. Fig 7 shows the clear relationship between the geological terranes, Alpine Fault, and bioavailable $^{87}Sr/^{86}Sr$, especially in the South Island. In particular, the model predicts the lowest bioavailable $^{87}Sr/^{86}Sr$ values (0.70570–0.70750) for the northwestern and southwestern portions of the South Island and Rakiura (Stewart Island). These rocks share a similar geological origin,

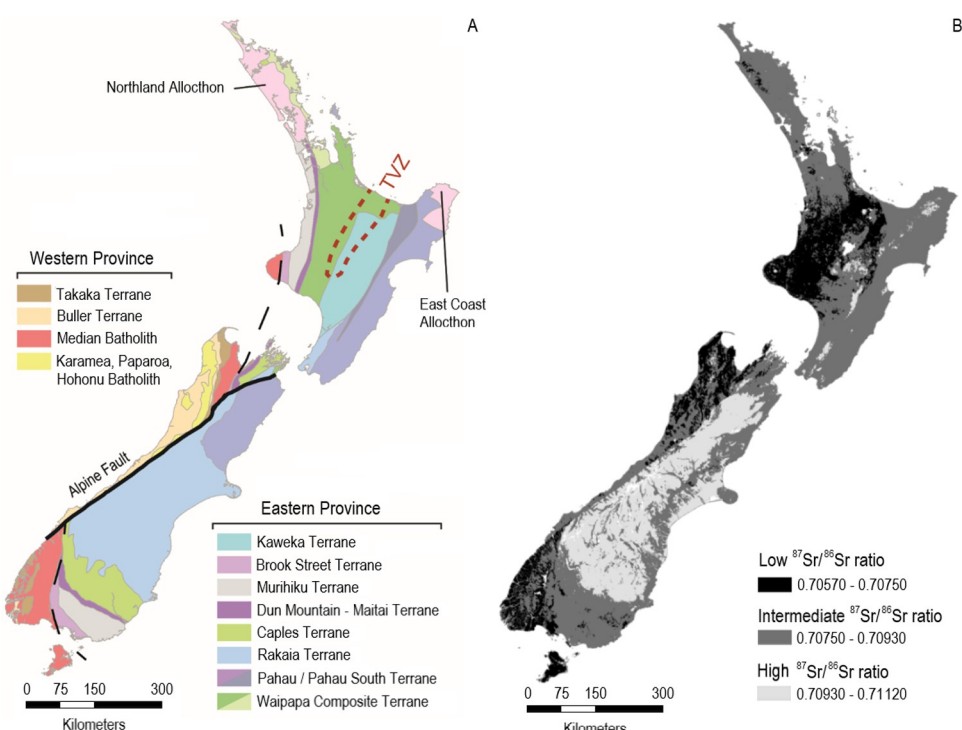

**Fig 7.** Diagrams showing simplified (A) basement terrane geology and (B) bioavailable $^{87}Sr/^{86}Sr$ isoscape highlighting regions with low, intermediate, and high $^{87}Sr/^{86}Sr$ values. The geology in (A) follows Edbrooke et al. [117]. The Alpine Fault is shown as a solid black line; the division between Eastern Province and Western Province basement rocks is shown by a black dashed line; and the Taupo Volcanic Zone (TVZ) outline is shown by a dotted red line. Low, intermediate, and high Sr value ranges for simplified isoscape (B) were determined using the "Equal Intervals" symbology function in ArcGIS Pro. Black denotes regions of low $^{87}Sr/^{86}Sr$ values (0.70570–0.70750), dark grey denotes intermediate $^{87}Sr/^{86}Sr$ values (0.70750–0.70930), and light grey denotes regions with high $^{87}Sr/^{86}Sr$ values (0.70930–0.71120). Figure developed in ArcGIS Pro using a coastlines feature layer (sourced from Natural Earth) and projected to NZTM 2000.

belonging to the Paleozoic Buller and Takaka terranes of the Indo-Australian plate. The rocks of these terranes are broadly comprised of metasediments and intruding plutonic igneous rocks, called batholiths, that formed along the previously active margin of Gondwana between 500–100 Ma [112,113]. Many of these rocks are of igneous origin or derivation, and therefore tend to have very low $^{87}Sr/^{86}Sr$ values despite their older age [114,115], though some plutonic igneous rocks are known to have higher $^{87}Sr/^{86}Sr$ values owing to their derivation from melting ancient crust [116].

The lowest predicted bioavailable $^{87}Sr/^{86}Sr$ isoscape values in the North Island also partially correspond with basement geology–some areas of low predicted $^{87}Sr/^{86}Sr$ isoscape values correspond with the Dun Mountain-Maitai and the Murihiku terranes (Fig 7). The Dun Mountain-Maitai Terrane is composed of ultramafic rocks overlaid by plutonic and volcanic sequences, that are covered by sedimentary deposits, while the Murihiku Terrane consists largely of sedimentary and volcanic materials [118]. However, in the central North Island bioavailable strontium ratios also appear to be heavily influenced by Quaternary volcanic deposits relating to eruptives from the Taupō Volcanic Zone (S2 Fig 1 in S2 File). These eruptive rocks include ignimbrites and volcanic ashes that date volcanism from around 1.6 Ma to present day [119]. Therefore, these deposits and the ashes that predate them are young and are expected to display low $^{87}Sr/^{86}Sr$ values. The area immediately surrounding Mount Taranaki in the westernmost region of the central North Island (Fig 7) has been active since c. 130 thousand years

ago (Ka) [120] and is also comprised of volcanic rocks that display the low $^{87}$Sr/$^{86}$Sr values characteristic of younger igneous formations [55]. The partial dependence plots (Fig 5) demonstrate this relationship between $^{87}$Sr/$^{86}$Sr values and age as well.

Conversely, the highest $^{87}$Sr/$^{86}$Sr values (0.70930–0.71120) of Aotearoa are predicted for areas comprised of the Mesozoic Caples, Rakaia, and Pahau terranes which stretch from the lower South Island along the eastern portion of the North Island (Fig 7). These terranes have mudstone/sandstone protoliths that are variably metamorphosed, but generally increase in metamorphic grade the closer they are to the Alpine fault [121,122]. The Caples Terrane consists of volcaniclastic feldspathic metasedimentary rocks [123]. The Rakaia and Pahau terranes contain variably metamorphosed quartzofeldspathic sedimentary rocks [124]. These rocks are overlain by aerially expansive alluvium and loess deposits in the central east portion of the South Island, around the Canterbury Plains (Fig 7 and S2 Fig 1 in S2 File). The elevated $^{87}$Sr/$^{86}$Sr values for these regions (Fig 7) most likely result from the more felsic (rich in silica) content of the rocks [64].

The juxtaposition of differing geologies along the Alpine Fault also has clear effects on the distribution of bioavailable $^{87}$Sr/$^{86}$Sr values in Aotearoa. This fault is primarily a strike-slip system, however uplift on the southeastern side of the fault has resulted in the formation of the Southern Alps. This mountain chain runs along the Alpine Fault and marks the distinction between high and intermediate $^{87}$Sr/$^{86}$Sr values in the model (Fig 7). The elevation partial dependence plot (Fig 5) shows that $^{87}$Sr/$^{86}$Sr values increase with increasing elevation. In addition, erosional forces associated with uplifted areas have generated massive amounts of gravel and dust that are older and have high $^{87}$Sr/$^{86}$Sr values. This gravel and dust have been transported by glaciers, glacier-fed streams, and prevailing westerly winds into the Canterbury plains to the east of the Alps [48,108,125], and therefore affecting $^{87}$Sr/$^{86}$Sr values in this region. Koffman et a. [48] note that the glacial activity and associated glacial outwash expanded the Canterbury Plains by approximately 30,000 km$^2$ by depositing sediments with $^{87}$Sr/$^{86}$Sr values ranging from 0.70950 to 0.71650. Glaciers to the west of the Southern Alps did not create similar outwash plains because they terminated near the edge of the continental shelf creating a distinction between high $^{87}$Sr/$^{86}$Sr values to the East and intermediate $^{87}$Sr/$^{86}$Sr values (0.70750–0.70930) in the West [48].

Although geological $^{87}$Sr/$^{86}$Sr affects bioavailable $^{87}$Sr/$^{86}$Sr, atmospheric variables clearly interact with geology to produce a complex isoscape. Despite its small size, Aotearoa's climate varies vastly with a warm subtropical climate in the Northland region, a cool temperate climate throughout most of the country, and alpine weather in the mountainous West Coast and Southland regions of the South Island [125]. The mean annual temperature plot (Fig 5) demonstrates that temperature has a significant effect on bioavailable $^{87}$Sr/$^{86}$Sr in Aotearoa where higher $^{87}$Sr/$^{86}$Sr values occur in areas with lower temperatures. This pattern correlates with geological and latitudinal differences between the North and South Islands. The former experiences warmer weather and has more volcanic and low-Sr lithologies (Fig 7), while the latter experiences colder weather and the highest $^{87}$Sr/$^{86}$Sr values occur along the Southern Alps and the Canterbury plains. Precipitation is another atmospheric variable that plays a key role in bioavailable $^{87}$Sr/$^{86}$Sr distributions. The Southern Alps experience some of the highest recordings for mean annual precipitation in Aotearoa, measuring up to 6,000 mm/year, while the land to their east records the lowest rainfall at 250–1,000 mm/year [125,126]. Increased precipitation on this coastal setting probably facilitates a high rate of marine aerosol deposition that is characterized by lower $^{87}$Sr/$^{86}$Sr signatures, as well as historically high deposition of volcanic ash from surrounding volcanic centers particularly in the northwestern South Island and the North Island [55]. This is reflected in the partial dependence plot for the sea salt aerosol deposition where higher rates of deposition are associated with lower $^{87}$Sr/$^{86}$Sr values (Fig 5).

However, the lack of bioavailable $^{87}Sr/^{86}Sr$ samples from these regions does limit the accuracy of the isoscape here (Fig 7). No cow milk samples were obtained from these regions.

## Testing the bioavailable isoscape using cow milk samples

Using the first bioavailable $^{87}Sr/^{86}Sr$ isoscape developed here for Aotearoa, this study sought to validate the use of strontium isoscapes for provenancing research, such as food product verification, by predicting the origin for cow milk sampled from around the country. Fig 6 illustrates that all cow milk samples, with one exception (Cow 7), fall within the 95% confidence interval of the line of best fit. This indicates that their measured $^{87}Sr/^{86}Sr$ values reflect the predicted values obtained from the bioavailable $^{87}Sr/^{86}Sr$ isoscape, illustrating that the bioavailable $^{87}Sr/^{86}Sr$ isoscape performs well and has the potential to be utilized in provenancing applications.

The relatively high regression residual for sample Cow 7 might be due to the addition of non-local feed to the cow's diet including health supplements, supplementary feed from other Aotearoa regions including ryegrasses (with high calcium content), imported palm kernel expeller (PKE), or even transportation of the cow between regions in Aotearoa that could introduce foreign $^{87}Sr/^{86}Sr$ values. However, dairy farms that contributed cow milk samples [97] had enforced a controlled feeding regime where all cattle were expected to be pasture-fed on-site and were not provided with supplementary feed options. In Aotearoa, livestock and raw cow's milk are commonly transported inter-regionally. Prior to going through the pasteurization and production processes, and depending on processing plant availability or maintenance, milk may be transported some distance, between the west and east coasts, particularly in the South Island. The outlier sample, Cow 7, may therefore represent a cow that 1) was not kept on a strict diet or 2) had been transported from another region in Aotearoa.

The accuracy and precision of predictions was measured as the distance from their known place of origin to the nearest predicted area of potential origin using different probability quantile thresholds (10%, 20%, and 33%). As expected, the top 33% threshold produced the most accurate (i.e., closest to known origin) predictions, but those predictions covered more potential areas and were less precise (Tables 1 and 2 and S4 File). On the other hand, the top 20% and top 10% thresholds provided more constrained potential region-of-origin predictions, making them more precise, but their accuracy decreased (Tables 1 and 2). Specifically, the average distance away from the place of known origin was 7.05 km for the top 33% threshold, 16.99 km for the top 20%, and 30.73 km for the top 10% threshold (Table 2). This showed that there was a trade-off between precision and accuracy for prediction outputs. In the Supplementary Information (S4 File), we provide prediction maps of the top 33% probability quantile for each cow milk sample.

In total, 73% (24 out of 33) of milk samples fell into the intermediate $^{87}Sr/^{86}Sr$ range with values between 0.70750 and 0.70930 (Figs 6 and 7). Milk that fell into this range originated from the Waikato, Canterbury, Southland, Northland, Manawatu-Wanganui, and Nelson regions (Figs 1 and 6). As indicated by the isoscape, there is high redundancy for bioavailable $^{87}Sr/^{86}Sr$ across Aotearoa. When samples had $^{87}Sr/^{86}Sr$ values that fell outside of this intermediate range, region-of-origin predictions were more likely to be very accurate and precise. For example, samples Cow 4 and Cow 5 had the lowest $^{87}Sr/^{86}Sr$ values (< 0.70750) and produced highly accurate and precise predictions for all probability quantile thresholds (Table 1, Fig 6 and S4 File). Furthermore, samples Cow 11–14, 16, 17, and 30 had $^{87}Sr/^{86}Sr$ values greater than 0.70930 and all predicted regions of potential origin less than 3 km away from their known place of origin for the top 33% and 20% thresholds (Table 1, Fig 6, and S4 File). When $^{87}Sr/^{86}Sr$ values did not fall into the low or high ranges, the accuracy of origin predictions

decreased, meaning that the distance away from the known place of origin tended to be larger because the assignment model predicted many potential regions of origin with similar bio-available $^{87}Sr/^{86}Sr$ values. In these cases, utilizing a second isotope system, such as $\delta^2H$ and $\delta^{18}O$, may help to constrain the predicted region-of-origin [31,37].

In terms of accuracy, Canterbury region cow milk samples (n = 8) produced the most accurate origin predictions across all probability thresholds with the smallest average distance between the known origin and the nearest predicted cell (Table 2 and S4 File). This strongly implies that the Canterbury cows have a local diet and do not have foreign feed introducing exogenous $^{87}Sr/^{86}Sr$ values. Conversely, Southland region milk samples (n = 11) produced the least accurate origin predictions across all thresholds (Tables 1 and 2). Fig 6 shows that the majority of the Southland cow milk samples have observed values below the 1:1 black line, implying that their actual $^{87}Sr/^{86}Sr$ values are higher than the predicted $^{87}Sr/^{86}Sr$ isoscape values. As mentioned previously, much of the South Island was glaciated about ~26,000 to ~18,000 BP and glacially fed rivers, specifically the Clutha and Mataura rivers and their tributaries, carried sediment from the Wakatipu Valley in the far western portion of the Otago region into the Southland and eastern Otago regions [48]. The fluvial transportation of sediments from the Wakatipu Valley may be a major contributor to the elevated $^{87}Sr/^{86}Sr$ values of the Southland cow milk samples. Furthermore, the low accuracy may result from the increased geologic diversity in Southland versus Canterbury, where the former is comprised of several terranes, ranging from mafic to felsic, and the former is dominated by the Rakaia Terrane (Fig 7).

Even though the Southland cow milk samples fell below the 1:1 line (Fig 6), many of the samples (Cow 19–23, 26, and 27) produced accurate predictions less than 5 km away from their known origin and three other Southland samples (Cow 24, 25, and 28) were only 13 to 19 km off their targets using the top 33% threshold (Table 1). One sample, Cow 29, had the highest $^{87}Sr/^{86}Sr$ value of the Southland group and was the least accurate where the nearest predicted cell was 86–123 km away from the cow's known place of origin (Table 1). Since the other milk samples from the Southland region produce accurate to moderately accurate predictions, we do not suspect that the elevated $^{87}Sr/^{86}Sr$ value of sample Cow 29 was due to its proximity to the coast (i.e., sea salt or dust aerosol deposition) or the introduction of foreign feed. Instead, it is likely that Cow 29 had been transported down to the Southland region from the Canterbury region which is more consistent with its elevated $^{87}Sr/^{86}Sr$ values. Milk samples were obtained through farms on a voluntary basis, so determining whether this sample had actually been transported from Canterbury was not possible.

## Further applications for provenancing materials in Aotearoa

Apart from food science applications, the bioavailable $^{87}Sr/^{86}Sr$ isoscape can assist with identifying the origin of new pests, unidentified materials, illegal agricultural products, and origin mislabeling. Specifically, $^{87}Sr/^{86}Sr$ can help biosecurity organizations within Aotearoa determine if a pest encountered post-border represents a newly introduced pest or an established population bearing local $^{87}Sr/^{86}Sr$ values [1,99,100,127,128]. For the key pastoral industry, Ferguson et al. [127] estimated that invasive invertebrates alone cost the sector between $1.7 to $2.3 billion NZD annually. New pests and diseases are introduced to Aotearoa through shipping and air traffic pathways [128–131]. One of the highest risk pests for Aotearoa is the exotic brown marmorated stink bug (*Halymorpha halys*) because it feeds on over 300 species of plants and infestation can ruin entire crops [7]. Recent attempts to define the origins of this invasive stink bug used $\delta^2H$ and $\delta^{18}O$ isotopes from their wings to determine whether a bug detected post-border represented a recently introduced foreign bug or an established population [2]. Holder et al. [2] concluded that the distinct $\delta^2H$ and $\delta^{18}O$ isotope signature of their

wings suggested a cooler climate origin, supporting evidence that the specimen was not from a locally breeding, southern hemisphere summer population. Currently, there are no rapid response tools to assess the provenance of these pests when they are introduced which limits the ability to develop effective measures to prevent their arrival in Aotearoa. This leaves agencies poorly informed as to the actual risk, with difficult decisions about embarking on expensive responses that may otherwise have to assume the presence of a locally established population.

Recently, Murphy et al. [63] developed a new mass spectrometric method to analyze $^{87}Sr/^{86}Sr$ in low-Sr samples, such as a single insect, in less time than traditional methods. This new analytical method enables this technology to be used within the limits for biosecurity, commanding very short time frames and availability of very little biological material. Now, coupled with this $^{87}Sr/^{86}Sr$ isoscape, well supported decisions based on probabilities can be made as to the likely Aotearoa or offshore origin of foreign pests, which is key to determine the most efficient response to prevent their establishment [2]. Prior to this point this had not been possible, anywhere. The identification and removal of foreign pests protects crops from foreign diseases and safeguards the future of endemic flora and fauna species throughout the country.

This $^{87}Sr/^{86}Sr$-focused approach can be applied globally to determine the local or foreign nature of pests on borders and ports of entry. However, this would require that areas of interest construct bioavailable $^{87}Sr/^{86}Sr$ isoscapes to compare their samples against. The Aotearoa $^{87}Sr/^{86}Sr$ isoscape produced by this study could be used alongside other lines of evidence (e.g., isotopic systems, chemical fingerprints), and strontium baseline models from other regions of the world to test whether "Made in New Zealand" food products display expected $^{87}Sr/^{86}Sr$ signatures or if they are fraudulent products.

Furthermore, law enforcement agencies could use the isoscape to predict the region of 'growing areas' or determine country of origin using bioavailable $^{87}Sr/^{86}Sr$ from confiscated illicit drugs, like marijuana or heroin [132,133]. Additionally, although Aotearoa does not encounter many unidentified human remains, if they were recovered and traditional identification methods failed to produce a positive identification, $^{87}Sr/^{86}Sr$ values from the unidentified person's teeth could help predict their region or country of origin [19,25,26,37,65,74,77,134,135].

The method of provenancing used in this study is useful because it provides a visual representation of the predicted probability surface that allows for ease of communication and comprehension. Though the predicted maps need to be generated by a specialist, once created, a non-specialist can interpret the prediction map and make decisions about where an unknown material may originate from within Aotearoa using their own criteria. The main limitation with applying the isoscape to provenancing investigations is that the predictions highlight many potential regions-of-origin that have similar $^{87}Sr/^{86}Sr$ values when using the top 33% threshold. The threshold can be set to 20% or 10% but, as demonstrated here, these thresholds reduce the accuracy of origin predictions. Instead, predicted areas could be further constrained by combining the predictive strength of $^{87}Sr/^{86}Sr$ with other isotope systems, such as $\delta^2H$ and $\delta^{18}O$ [37,74,89]. Isoscapes for $\delta^2H$ and $\delta^{18}O$ already exist for Aotearoa and can easily be combined in a dual-isotope approach to predict the region-of-origin for unknown materials using the built-in assignment features of the *assignR* package [104,136–138].

## Conclusion

As people, animals, and materials are transported across increasingly large distances in a globalized world, issues with biosecurity and food security are rising. The global pandemic of

COVID-19 has highlighted the difficulties we face in tracing and controlling the origins of animals and products that can transport viral loads. In Aotearoa, there are many endemic species and a strong local agricultural industry that must be protected from biosecurity threats. There is an urgent need to have tools which enable the provenancing of pests and agricultural products arriving in and being transported around the islands of Aotearoa, as well as confirm origins of products advertised as 'New Zealand made' that enter into overseas markets to protect valuable commodities from fraud. Isotopes are ubiquitous markers of provenance that are increasingly used to trace the origin of food or animals. In this study, we introduced the first bioavailable $^{87}$Sr/$^{86}$Sr isoscape for Aotearoa and demonstrated how the isoscape can be used to certify the origin of agricultural products. We improved upon existing methodology to develop a bioavailable $^{87}$Sr/$^{86}$Sr isoscape using the best available geospatial datasets to tune the regional isoscape. As anticipated, the primary drivers of bioavailable $^{87}$Sr/$^{86}$Sr variability are the underlying geology, soil pH, and aeolian (dust and sea salt) deposits. We then tested and proved that there is potential for utilizing $^{87}$Sr/$^{86}$Sr isotopes to determine the origin of cow milk and other agricultural products in Aotearoa.

Currently, there is little to no geo-referenced bioavailable $^{87}$Sr/$^{86}$Sr data derived from plants, animals, or soil in Aotearoa, beyond this study. As more data become available, the bioavailable $^{87}$Sr/$^{86}$Sr isoscape model can be recalibrated and further improved, though we do acknowledge that sampling to fill current geographic gaps in the model is a costly undertaking. With the availability of this baseline bioavailable $^{87}$Sr/$^{86}$Sr isoscape, our hope is to promote further research using $^{87}$Sr/$^{86}$Sr isotopes in Aotearoa and abroad.

Current provenancing methods in Aotearoa rely heavily on trace elements and light stable isotopes ($\delta^2$H, $\delta^{15}$N, $\delta^{18}$O, $\delta^{13}$C) to predict region-of-origin. The information provided by light isotopes and trace elements are useful for excluding potential regions of origin from consideration, but predictions can be further constrained if combined with other geologically derived isotope systems, like $^{87}$Sr/$^{86}$Sr [37,89]. Future provenancing projects in Aotearoa should implement a multi-isotope approach that uses $^{87}$Sr/$^{86}$Sr alongside other light stable isotopes to produce both accurate and precise region-of-origin predictions.

## Supporting information

**S1 File. Sample preparation and analysis.** S1.1. Aotearoa Plant Sample Preparation. S1.2. Aotearoa Topsoil Sample Preparation. S1.3. Collected Plant and Topsoil MC-ICP-MS Analysis. S1.4. Additional Plant and Topsoil Preparation and Analysis. S1.5. Cow Milk Sample Preparation and Analysis.
(DOCX)

**S2 File. Random forest model and auxiliary variables. S2 Table 1. Geological and climatic variables used in random forest regression.** D = discrete variable; C = continuous variable. **S2 Table 2. "Toprock" Category (#1–67) Descriptions. S2 Fig** 1. The top performing covariate "Toprock" selected by VSURF package. The 67 toprock principal surface lithology types have been simplified but all types are detailed in S2 Table 2. Data sourced from LRIS (https://lris.scinfo.org.nz/layer/48065-nzlri-rock/; 2010) [S5.18] and GNS (fault lines shapefile) [S5.20]. The "toprock" shapefile does not include data for Rakiura (Stewart Island) or Wharekauri (Chatham Islands).
(DOCX)

**S3 File. Bioavailable 87Sr/86Sr Isoscape–Color Version. S3 Fig 1. Bioavailable 87Sr/86Sr isoscape (color version).** The highest $^{87}$Sr/$^{86}$Sr ratios are depicted in blue and the lowest are shown in purple. The bioavailable $^{87}$Sr/$^{86}$Sr isoscape ($R^2$ = 0.53, RMSE = 0.00098)

demonstrates the predicted $^{87}$Sr/$^{86}$Sr values, ranging from 0.70567 to 0.71118, for the entire country including the Chatham Islands. Figure developed in ArcGIS Pro using a coastlines feature layer (sourced from Natural Earth) and projected to NZTM 2000.
(PDF)

**S4 File. Aggregated 87Sr/86Sr data.** Excel = "S4_BioSrData".
(XLSX)

**S5 File. Cow milk sample predictions.** PDF = "S5_CowMilkSamplePredictions". **Cow Milk Sample Predictions—Top 33% Posterior Probabilities.** Regions highlighted in blue represent the top 33% of areas predicted as potential regions-of-origin using bioavailable $^{87}$Sr/$^{86}$Sr where values "1" = Likely Origin and "2" = Unlikely Origin. The box in top left shows the area surrounding the cow milk sample's actual place-of-origin marked by yellow circle. The hillshade layer was created using a GNS DEM 8m shapefile.
(PDF)

**S6 File. References.**
(DOCX)

## Acknowledgments

A special thanks to the Centre for Trace Elements team at the University of Otago for the countless hours of lab guidance and supervision. Additional thanks to AsureQuality for assistance collecting B3 samples and to David Murphy, Queensland University of Technology for managing analysis of the B3 samples. We acknowledge the SPATIAL (Spatio-temporal Isotope Analytics Lab, University of Utah) Short Course for providing the environment to design this work and the funds to carry it out to RTK.

## Author Contributions

**Conceptualization:** R. T. Kramer, R. L. Kinaston, P. W. Holder, K. F. Armstrong, C. L. King, C. P. Bataille.

**Data curation:** R. T. Kramer, R. L. Kinaston, P. W. Holder, K. F. Armstrong, W. D. K. Sipple, A. P. Martin, G. Pradel, R. E. Turnbull, K. M. Rogers, M. Reid, D. Barr, K. G. Wijenayake.

**Formal analysis:** R. T. Kramer, P. W. Holder, K. F. Armstrong, M. Reid, D. Barr, K. G. Wijenayake, C. H. Stirling, C. P. Bataille.

**Funding acquisition:** R. T. Kramer, R. L. Kinaston, P. W. Holder, K. F. Armstrong, C. L. King, H. R. Buckley, C. P. Bataille.

**Investigation:** R. T. Kramer, P. W. Holder, K. F. Armstrong, C. L. King, C. P. Bataille.

**Methodology:** R. T. Kramer, D. Barr, C. H. Stirling, C. P. Bataille.

**Project administration:** R. T. Kramer, R. L. Kinaston, C. L. King, H. R. Buckley, C. P. Bataille.

**Resources:** R. T. Kramer, C. H. Stirling, C. P. Bataille.

**Software:** R. T. Kramer, M. Reid, D. Barr, C. H. Stirling, C. P. Bataille.

**Supervision:** R. L. Kinaston, C. L. King, H. R. Buckley, C. P. Bataille.

**Validation:** R. T. Kramer, C. P. Bataille.

**Visualization:** R. T. Kramer, C. P. Bataille.

**Writing – original draft:** R. T. Kramer, C. P. Bataille.

**Writing – review & editing:** R. T. Kramer, R. L. Kinaston, P. W. Holder, K. F. Armstrong, C. L. King, A. P. Martin, G. Pradel, R. E. Turnbull, K. M. Rogers, H. R. Buckley, C. H. Stirling, C. P. Bataille.

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
