## [Decision Letter · Decision Letter 0]

16 Nov 2021

PONE-D-21-33406A bioavailable strontium (87Sr/86Sr) isoscape for Aotearoa New Zealand: Implications for food forensics and biosecurityPLOS ONE

Dear Ms Kramer,

Thank you for submitting your manuscript to PLOS ONE. I have now received two reviews and I am pleased that both reviewers find the manuscript to be of high quality and worthy of publication. There are some changes that both of the reviewers suggest---Reviewer 2 in particular provides detailed comments and recommends some re-structuring of the text, includes some additional references that would be helpful to include, and also poses some useful questions about how provenance can be applied in the use of strontium isotopes based upon your data from Aotearoa New Zealand. So I would ask that you study the reviewer comments and I invite you to submit a revised version of the manuscript that addresses the points raised during the review process.

We look forward to receiving your revised manuscript.

Kind regards,

Lee W Cooper, Ph.D.

Section Editor, Biogeochemistry

PLOS ONE

Journal Requirements:

2. In your Methods section, please provide additional information regarding the permits you obtained to collect samples for the present study. Please ensure you have included the full name of the authority that approved the field site access and, if no permits were required, a brief statement explaining why.

3. We note that Figures 1, 3, 7 and S1 in your submission contain [map/satellite] images which may be copyrighted. All PLOS content is published under the Creative Commons Attribution License (CC BY 4.0), which means that the manuscript, images, and Supporting Information files will be freely available online, and any third party is permitted to access, download, copy, distribute, and use these materials in any way, even commercially, with proper attribution. For these reasons, we cannot publish previously copyrighted maps or satellite images created using proprietary data, such as Google software (Google Maps, Street View, and Earth). For more information, see our copyright guidelines: http://journals.plos.org/plosone/s/licenses-and-copyright.

a. You may seek permission from the original copyright holder of Figures 1, 3, 7 and S1 to publish the content specifically under the CC BY 4.0 license.  

Reviewers' comments:

Reviewer's Responses to Questions

**Comments to the Author**

1. Is the manuscript technically sound, and do the data support the conclusions?

Reviewer #1: Yes

Reviewer #2: Yes

2. Has the statistical analysis been performed appropriately and rigorously? 

Reviewer #1: Yes

Reviewer #2: Yes

3. Have the authors made all data underlying the findings in their manuscript fully available?

Reviewer #1: Yes

Reviewer #2: Yes

4. Is the manuscript presented in an intelligible fashion and written in standard English?

Reviewer #1: Yes

Reviewer #2: Yes

5. Review Comments to the Author

Reviewer #1: This is a very well written manuscript and very well performed study. I have only a handful of minor comments:

1. Line 59: the "18O" font looks strange.

2. Lines 526-528: What is the mechanism by which temperature influences bioavailable Sr isotopes?

3. The word "data" is plural is "becomes" should be "become".

Reviewer #2: The authors provide a very well done and much needed contribution to the strontium isotope global literature in the form of a predicted isoscape with validation from Aotearoa. The production of the isoscape is rigorous and follows known best practices in the literature. Most of my comments are relatively minor. My one major question comes with the applicability sourcing using the 33% predictive threshold; while most milk samples did have a nearby highly probably source, the highest density of probable source locations was often on the other side of the country (ex: Cow 19-21)! How can this predictive model be used in a way that is meaningful to unknown samples? I go into more detail on these question in the line-by-line comments below.

Overall this is a very strong paper and a dataset worthy of publication. I thank the authors for their diligence and look forward to their revisions.

----

Line 32: “Bioavailable stable and radiogenic…” – This should just read “radiogenic.” There are no stable isotopes of strontium. Also the word ‘ubiquitous’ feels strange here, and I’m not sure what meaning the authors are trying to convey with it. Consider selecting a different word.

Lines 52-68: This goes fairly in-depth right off the bat about using isotopes to track the origins of pest insects themselves; while interesting, this is not actually relevant to the focus of the paper, which tests the validity of the isoscape to source products. The discussion of the application to identifying the imported vs “homegrown” nature of pest insects is fine, but belongs in the Discussion section and not as the opening to the paper. You could also move the discussion of the isotopic application into the ‘previous research’ summary beginning on line 82.

111 - Hydrogen and oxygen are used in many places to determine provenance within a regional scale. This has even been done with milk (Vieira Silva et al 2013, Boito et al 2021, links to journals below). The authors are not incorrect about the ambiguity surrounding all the inputs into an oxygen isotope ratio, but that does not mean that it cannot be used at a regional scale. Like strontium, it all depends on the variation present and the patterning of that variation!

https://analyticalsciencejournals.onlinelibrary.wiley.com/doi/pdfdirect/10.1002/rcm.9160

https://www.sciencedirect.com/science/article/pii/S0958694613002835?casa_token=ylfxor7hGakAAAAA:jgJdG2Raca2CaCmPbbh2-zwbFpWeo4NiEC_LfMPXJO-mh_9y6kSdQeHD5OnfQn_R5GDHbzsK

129-131 – isn’t this same limitation true of strontium isotope ratios? Without baseline knowledge – something your isoscape is providing here! – you are unable to draw conclusions about provenance.

172 – Isoscapes for Tanzania and Kenya should be included here as well (Janzen et al 2020):

https://www.sciencedirect.com/science/article/pii/S0031018220304028?casa_token=d7oU3el2pFEAAAAA:pe1lNFHYhVvwUZAYUi_CJT5r9WnHMmNglGWUoVxAnjxEol-kYVQDWOz94mTh5S37xOk1KLnq

199 – Riparian areas are also shown to influence the values of plants growing near mobile waterways because of the variable values from that water flowing over different isotopic zones (Sillen et al 1998, Hamilton et al 2019). This should also be included in the discussion of possible influences on the isotopic ratio, and discussed in the context of the authors’ collection of samples (were any samples in riparian zones? Could this influence results?)

https://www.sciencedirect.com/science/article/pii/S0016703798001823?casa_token=7nWAfydRCjEAAAAA:ihiJezZaSMJUn-CbsWEdghHuUkAD2WY7f13ZK6ZsisCpn5s8aNKDW34gmqkbwlD1D4ZkYCiv

https://onlinelibrary.wiley.com/doi/abs/10.1002/ajpa.23932?casa_token=HXpMWQKHv14AAAAA%3AtCW1VHUbObhoDRlHrkqWv3G435VCxxgsubp89JCXlgVA6Fr_K4aoTkeNIxpYgTpIsS0isyUFquS12g

206 – were samples collected far enough from roads to avoid contamination from dust from vehicles, etc? Also, include information about topical cleaning of samples (were they brushed clean or washed before being dehydrated, for example) or other precautions taken to avoid dust contamination in the measurement.

290 – what is the turnover time for milk production? In other words, how long would cows need to be feeding locally to ensure that the signal within the milk produced is local?

368 – how can the categorical variable of “principal surface lithology type” be inversely correlated with something? How can a categorical variable increase or decrease?

Line 400 / Fig 6 – Can you explicitly discuss the source the predicted Sr/Sr value in the paper body? You describe it well in the figure 6 caption as being based on the isoscape and known area of sample collection, but the way it reads in the paper makes it sound like it is a value related to the posterior probabily map somehow.

404 – Line 404 – remove comma after “all”

Here is my major question with this particular study. I appreciate the methodology used here, and the assignR function is an excellent analytical choice. However, looking at the provided maps, I have no idea how this information would be useful for an unknown sample in providing provenience information. The metric of “distance to the nearest predicted origin” feels somewhat misleading, given how many high-probability cells there are all over the map, often at much greater densities far away from the true point of origin. Given this, how would this method actually be useful to determining the validity of a product’s stated origin? You discuss that the accuracy goes down while precision goes up using smaller threshholds, but isn’t that exactly what you would expect? Without the precision, how is this technique useful in practice?

427 – Is the accuracy/precision of estimates related to the general heterogeneity of the area? I would image the more heterogenous (geologically) the area is, the more precise the estimate might be, whereas more homogenous areas might have higher accuracy with less precision. Do you see any correlations like this?

526 and 590 – Is there any proposal causal mechanism in the literature to explain MAT impacting strontium ratio? I have never seen this proposed, and can’t think of a reason why temperature alone would change the isotopic composition of an area. It reads here as causal (MAT causes Sr/Sr to change) – is that accurate, or is it merely a correlation observed (because of interaction effects with elevation, age, lithology, etc)? Please be clear.

620 – this would be a good place for your discussion of pests from the introduction!

General comments:

Is there a specific area that counterfeit products tend to come from? Are there any published isotopic values from these areas?

I would love to see a color version of Fig 3!

6. PLOS authors have the option to publish the peer review history of their article (what does this mean?). If published, this will include your full peer review and any attached files.

Reviewer #1: No

Reviewer #2: No

---

## [Author Response · Author response to Decision Letter 0]

8 Jan 2022

Please see uploaded "Response to Reviewers" document included with the other revised materials. All comments have been responded to.

---

## [Editor Report · Decision Letter 1]

11 Feb 2022

A bioavailable strontium (87Sr/86Sr) isoscape for Aotearoa New Zealand: Implications for food forensics and biosecurity

PONE-D-21-33406R1

Dear  Robyn,

Thank you for re-submitting your manuscript. I am not sure why it took some time to get back to me since you submitted it at the end of December, but I've had a chance to look at the changes you have made in response to the reviewer recommendations. I think you have more than satisfied the needs to improve the manuscript in response to the reviewer recommendations, and I'm  pleased to inform you that your manuscript has been judged scientifically suitable for publication. It will be formally accepted for publication once it meets any outstanding technical requirements as judged by the editorial office.

Within one week, you’ll receive an e-mail detailing any additional required amendments that are judged necessary by the editorial office. When these have been addressed, you’ll receive a formal acceptance letter and your manuscript will be scheduled for publication.

If your institution or institutions have a press office, please notify them about your upcoming paper to help maximize its impact. I have been told that PLOS staff (Natasha MacDonald; nmcdonald@plos.org) is interested in possibly helping with publicizing your results upon publication. If your institution will be preparing press materials, please inform our press team as soon as possible -- no later than 48 hours after receiving the formal acceptance. Your manuscript will remain under strict press embargo until 14:00 USA Eastern Time on the date of publication. For more information, please contact onepress@plos.org.

Thank you again for your efforts to present your results in PLOS ONE.

Kind regards,

Lee W Cooper, Ph.D.

Section Editor

PLOS ONE

---

## [Editor Report · Acceptance letter]

17 Feb 2022

PONE-D-21-33406R1 

A bioavailable strontium (87Sr/86Sr) isoscape for Aotearoa New Zealand: Implications for food forensics and biosecurity 

Dear Dr. Kramer:

I'm pleased to inform you that your manuscript has been deemed suitable for publication in PLOS ONE. Congratulations! Your manuscript is now with our production department. 

Kind regards, 

on behalf of

Dr. Lee W Cooper 

Section Editor

PLOS ONE